# Features and Management of New Daily Persistent Headache in Developmental-Age Patients

**DOI:** 10.3390/diagnostics11030385

**Published:** 2021-02-24

**Authors:** Laura Papetti, Giorgia Sforza, Samuela Tarantino, Romina Moavero, Claudia Ruscitto, Fabiana Ursitti, Michela Ada Noris Ferilli, Federico Vigevano, Massimiliano Valeriani

**Affiliations:** 1Department of Neuroscience, Paediatric Headache Center, Bambino Gesù Children Hospital, 00165 Rome, Italy; laura.papetti@opbg.net (L.P.); giorgia.sforza@opbg.net (G.S.); samuela.tarantino@opbg.net (S.T.); romina.moavero@opbg.net (R.M.); fabiana.ursitti@opbg.net (F.U.); michela.ferilli@opbg.net (M.A.N.F.); 2Child Neurology Unit, Systems Medicine Department, Tor Vergata University Hospital of Rome, 00133 Rome, Italy; claudia.ruscitto@opbg.net; 3Department of Neuroscience, Neurology Unit, Bambino Gesù Children Hospital, 00165 Rome, Italy; federico.vigevano@opbg.net; 4Center for Sensory-Motor Interaction, Aalborg University, 9220 Aalborg, Denmark

**Keywords:** new daily persistent headache, children, chronic headache, NDPH

## Abstract

Introduction. Our aim was to investigate the clinical features of primary new daily persistent headache (NDPH) in a cohort of paediatric patients. Methods. We reviewed the data of patients with persistent daily headache, attending the Headache Centre of Bambino Gesù Children from the January 2009. The ICHD-III criteria were used for diagnosis. Statistical analysis was conducted to study possible correlations between NDPH and population features (age and sex), NDPH and headache qualitative features, and NDPH and response to pharmacological therapies. Results. We included 46 subjects with NDPH. The features of pain more closely resembled those of migraine than to those of tension-type headache (62 vs. 38%). The NDPH patients showed nausea and vomiting less frequently than migraine ones (28.6 vs. 48.2%, *p* < 0.01). A total of 75% of NDPH patients experienced an onset of the symptoms in the winter months (November to February) (*p* < 0.01). NDPH was less common in very young children under 10 years of age. Almost 58% of NDPH patients received pharmacological therapy and the most used drug was amitriptyline. A reduction of attacks by at least 50% in a month was detected in 30.6% of patients. Conclusions. NDPH can be very disabling and correlates with seasonal factors. Although long term pharmacological therapy is recommended, considering the long duration that this headache can have, there are no data supporting the treatment choice.

## 1. Introduction

New Daily Persistent Headache (NDPH) consists of a persistent headache, daily from its onset, which is clearly remembered [1]. NDPH prevalence ranges from 0.03 to 0.1% in the general population and is higher in children and adolescents than in adults [2]. NDPH accounts for 1.7 to 10.8% of children and adolescents with chronic daily headache [3].

NDPH is very often disabling and may significantly affect the individual’s quality of life. It can even lead to psychiatric conditions [2,3]. The mechanisms that cause continuous headache typical of NDPH are not well-understood. In the literature, there are clinical studies conducted on the adult population that describe the characteristics of pain and the presence of any trigger factors [2,3]. Far fewer are the paediatric pooling studies with NDPH [4,5]. Furthermore, especially for what concerns the therapeutic management of NDPH, to date, we have conclusive information. Although therapeutic controlled drug trials are lacking, some authors [2] have suggested that NDPH should be treated according to the predominant headache phenotype; therefore, prophylactic drugs are generally used for high-frequency migraine or tension-type headache. However, the need for pharmacological treatment of NDPH at developmental age is quite debated considering the existence of a self-limiting subtype that typically resolves within several months without therapy [1].

The aim of this retrospective study was to describe the clinical features, trigger factors, and therapeutic options of NDPH in a paediatric cohort to better understand this enigmatic disorder.

## 2. Materials and Methods

We retrospectively reviewed the charts of patients attending to the tertiary paediatric medical Headache Center of Bambino Gesù Children’s Hospital. The electronic database of the headache clinics was searched for all children and adolescents up to 18 years of age, diagnosed with continuous primary headache with acute onset during the 2013 to 2019-time interval. To enter in the analysis, the main inclusion criteria was a persistent headache, daily from its onset, and present for more than three months. Exclusion criteria were a prior headache history, non-continuous remitting headache, secondary causes of headache, the presence of other internist and/or neurological illness and/or head or neck trauma, and the absence of a follow-up visit at least three months after the first observation.

Patients with a previous history of headache that could therefore be defined as chronic migraine (CM) or chronic tension-type headache were excluded from the analysis. Trigeminal autonomic cephalalgias (TACs) were excluded based on the absence of specific criteria such as autonomic symptoms or response to indomethacin trial [1].

Secondary headaches were appropriately excluded in all patients through medical history, physical examination (including fundus oculi examination), contrast-enhanced magnetic resonance imaging (MRI), and, in some cases, lumbar puncture with cerebrospinal fluid (CSF) pressure assessment. MRI was considered as a mandatory inclusion criterion. A lack of lumbar puncture was not considered an exclusion criterion unless there were clinical or neuroradiological elements suggesting a CSF pressure disorder (Figure 1). Secondary causes were identified in 11 patients with headaches resembling NDPH (23%) and included idiopathic intracranial hypertension (4/60), Chiari 1 syndrome (2/60), and brain tumors (5/60). In three patients eligible for the diagnosis of NDPH, the ICHD III criteria for hemicrania continua were satisfied. In particular, in these patients, a trial with indomethacin confirmed the diagnosis. The remaining 46 patients (12% of the whole CDH population) received a definitive diagnosis of NDPH according to the ICHD-III criteria (Figure 1). None of the definitive NDPH patients had a previous history of headache.

Data on demographics, headache symptoms, and other clinical headache-related parameters were collected from the medical files of the patients who were found eligible to be included in the study. Electronic medical records included the following information: demographic data (age, sex), familiar medical history including headaches, pregnancy and birth history, past medical history, anthropometrical data (weight and height), general physical exam and neurological exam including fundus oculi. Medical charts also included the results of possible neuroimaging exams and the data from headache diaries. Headache diaries report the number of attacks per month, the duration of the attacks, qualitative features of pain, presence of associated symptoms (nausea, vomiting, phonophobia, and photophobia), intensity of pain, name of drug for the attack and response to therapy for the attack. Psychiatric scales were not systematically used in the assessment.

Patients were divided into four age groups: 0–6, 7–10, 11–14, and 15–18 years.

Clinical data concerning the duration and qualitative features of the headache attacks, related symptoms, and pharmacologic therapies to induce remission were issued from the first and follow-up visits. These data were collected from interviews with children and/or their parents. For very young children, the headache frequency and symptoms were determined by the child’s complaints and the parents’ impression from the child’s behaviour (according to the ICHD-III criteria) [1]. In addition, parents were questioned about possible medication overuse of their child. The medical interview was always followed by a complete full physical and neurological examination of the patient. As the NDPH criteria included a distinct and clearly remembered onset, with pain becoming continuous and unremitting within 24 h (criterion B) [1], we looked for the possible presence of trigger factors in the headache history of the patients. Written informed consent was obtained from the parents of the participants in this study. The study was approved by the Ethical Committee of Bambino Gesù Children’s Hospital.

Statistical analysis was conducted by SPPS version 22.0 and χ2 test was used to verify the dependence between categorical variables. We tested in detail whether there was an association between: (a) NDPH frequency and age category or sex of patients, (b) NDPH and headache qualitative features (nausea, vomiting, phonophobia, and photophobia), and (c) the use of pharmacological therapy (amitriptyline, topiramate, flunarizine, and L-5 hydroxytryptophan) and a reduction of at least 50% of the monthly days with a headache. Furthermore, a multiple-regression logistic analysis has been used to evaluate whether age at first attack (0–6, 7–10, 11–14, 15–18 years), history of trigger or type of pharmacological treatment (topiramate, 5-hydroxytryptophan, flunarizine, amitriptyline) influenced the outcome (continuous headache for more than six months). The outcome was selected as a dependent variable, and then all the other variables were tested as independent variables in a block entry to evaluate the *β* standardized coefficient, the standard error (SE), the significance and the upper and lower limit in a confidence interval (CI) of 95%.

A *p*-value of ≤0.05 was considered significant. A post-hoc correction for multiple comparisons was made using the Bonferroni’s test, setting the significance cut-off at a/n with α = 0.05 and final value of *p* = 0.01.

## 3. Results

### 3.1. Clinical Characteristics of NDPH Patients

According to the last version of ICHD, 60 patients (54% female and 46% male) had an onset of headache suggestive of possible NDPH (Figure 1). The mean age of selected patients was 13.4 years ± 3.2 standard deviation (SD) (range 5.3 to 17.9 years).

Among the NDPH patients, there was no difference between genders (55% females and 45% males, *p* > 0.05). NDPH was less frequent under six years of age (8.7%) and above 14 years of age (19.6%) while a significant higher prevalence of NDPH was found in the age groups between 7 and 10 years (28.3%) and between 11 and 14 years (43.5%) (*p* < 0.01).

Given the retrospective nature of this work we found that 15% of patients with possible NDPH were initially diagnosed at the first medical visit as chronic migraine or chronic tension-type headache. The reason for this was because a diagnosis was made solely based on the qualitative characteristics of the pain, without considering the de novo mode of the onset of the headache.

### 3.2. Triggering Events

In 66% of NDPH cases, the onset of headache was correlated with the patient experiencing a temporally limited event, while in the remaining 34% of cases, neither the patient nor his/her parents had memory of a triggering event.

Two types of triggering events were recognized: a stressful increase in school activities was referred to in 46% of cases, while, according to 22% of cases, an infection of the upper respiratory tract would have triggered headache. However, these conditions could not be classified as secondary headaches, e.g., headache secondary to infections, because pain persisted almost three months after the event (school-related stress and infection) was over. Both trigger factors correlated with our finding of a far higher NDPH incidence in the winter months, with a peak in January. During summer, a lower peak is recorded in mid-June, coinciding with the end of the school year (Figure 2).

### 3.3. Qualitative Features and Associated Symptoms

The qualitative characteristics of pain reminded those of migraine in 73% of NDPH patients and those of tension-type headache in 27% of cases. Pain location, quality, and intensity are resumed in Table 1. 

We found no statistically significant differences between males and females. Regarding the various age groups, the only difference was in the intensity of the pain which was more frequently mild in patients under the age of six years (75%) than in the other age groups (47% for 7–10 years; 42% for 11–15 years and 38% for 15–18).

Patients with chronic migraine (247/369 patients) were then compared with those with NDPH (46/369) to verify if there were differences in the associated sensorial and vegetative symptoms. We did not find a statistically significant difference between the proportion of patients who experience and do not experience photophobia and phonophobia between the group of CM (25% without photophobia, 74.5% with photophobia, 20.8% without phonophobia, and 79.2% with phonophobia; *p* > 0.05) and NDPH (20.8% without photophobia, 79.2% with photophobia, 37% without phonophobia, and 63% with phonophobia; *p* > 0.05) (Figure 3).

Regarding vegetative symptoms, NDPH patients experienced nausea and vomiting less often than CM patients (28.3% vs. 58%; *p* < 0.01).

### 3.4. Treatment

Data concerning the use of pharmacological therapy aimed to induce remission of pain were issued to 33 patients (67%). The remaining 16 patients (33%) did not undergo preventive drug treatment. In particular, 10 patients did not receive any preventive therapy, while the remaining six patients were treated with non-pharmacological therapies (three patients underwent nutraceutics, two underwent psychotherapy, and one acupuncture). The main reasons for the lack of drug therapy were the mild intensity of the attacks, the interference with school and extracurricular activities, and the refusal of drugs by parents. 

The drugs used included flunarizine (5 mg/die) (3.7%), topiramate (1–2 mg/kg/die) (14.8%), and amitriptyline (1 mg/kg/die) (92.3%).

Almost 80% of patients had a beneficial response (reduction in the frequency of attacks by at least 50%), while 20% of patients showed no improvement. We found that 35% of patients did not have a response to the attack drugs (nonsteroidal anti-inflammatory drugs or triptans). Only one patient had a concomitant history of medication overuse headache (MOH); after correcting the excessive use of drugs for the attack, we did not observe a change in the frequency of the headache.

### 3.5. Outcome

As for the outcome, the average duration of continuous daily pain from the onset was 8.4 ± 2 months. We observed also that 43% of patients had continuous pain resolution within six months, 39% within 12 months, and 18% kept referring to continuous pain after 12 months. After a mean follow-up period of 22 ± 5 months, we observed that, over time, 54% of the NDPH population returned to a remitting form of headache (complete remission or frequency of attacks less than five days per month), 35% a had history of relapsing remitting headache (periods of continuous daily headache alternating with pain-free periods), and 11% continued to have persistent headache from the onset.

Unfavourable outcome (persistence of pain after six months) predictors were no history of a trigger event (β 3.5; SD 1.0; CI 0.6–4.6) and no pharmacological treatment (β 2.6; SD 1.5; CI 0.3–5.1). 

## 4. Discussion

The recent study showed that, in NDPH patients, characteristics of pain and associated symptoms more closely resembled those of migraine than to those of tension-type headache. Triggers identified by patients and their families included stressful school activities or infectious events. The duration of the NDPH could persist for six months or more, and the worst outcome was associated with the absence of identifiable trigger factors and a lack of pharmacological therapy.

Moreover, this study underlines how some aspects of the ICHD-III diagnostic criteria need to be clarified in order to help the diagnosis of NDPH in developmental-age patients.

In the ICHD-III criteria, no reference is made to the specific qualitative characteristics of pain or the presence of associated symptoms to make the diagnosis of NDPH [1]. In our cohort, 73% of patients reported headache characteristics similar to those of migraine. This is consistent with prior studies showing that paediatric patients with NDPH have a migrainous phenotype of their headache [4]. However, when compared to patients with chronic migraine, our NDPH patients were less often referred with nausea and vomiting. Recently, a large study conducted on 1170 young people (ages 3 to 17) with continuous headache analyzed the main differences between CM and NDPH in paediatric-age patients [5]. Paediatric patients with NDPH compared to CM patients generally experienced onset at a later age, less often presenting with photophobia, and less frequently developing an MOH [5].

These studies, as well as ours, show that although the characteristics of pain resemble those of migraine, there are elements of distinction. Furthermore, we agree that what most characterizes NDPH and differentiates it from other chronic headaches is more the modality of onset than the characteristics of pain. This also makes it possible to distinguish patients with NDPH from patients with migraine or tension-type headache whose headaches have begun to change from episodic to chronic. None of our patients with NDPH had a previous history of headache.

Another important finding is that the role of trigger factors in NDPH should be more emphasized even at paediatric age. Patients with NDPH often have a very distinct memory of the onset of the headache and can relate it to a trigger event [1]. Trigger factors described in adults are very different from those in children with NDPH. Studies on adults have reported surgery, dental extractions, electric shock, anaesthesia, infections, psychological stress (work, family, or financial problems), lifestyle changes, or moderate trauma as being among the most frequent events [5,6]. Other reported precipitating factors include withdrawal from SSRIs, human papilloma virus vaccination, menarche and postpartum state, hormone manipulation with progesterone, toxin and medication exposure, cervical massage treatment, simple syncopal attack, and thyroid diseases [1,7].

An important datum is that some authors have pointed out that patients are often unaware of situations that can trigger headaches [8]. In the suspicion of NDPH it is therefore very important, when collecting the anamnesis, to question the patient about trigger events that have preceded the onset of pain. 

In a study of 40 paediatric headache patients with NDPH, precipitating events were noted in 88% of cases: febrile illness in 43%, preceding minor head injury in 23%, and cranial or extra cranial surgery in 10% [9]. Moreover, at paediatric age, chronic headaches often show a seasonal cyclical pattern, which has been related to school activities [10,11,12]. In paediatric NDPH, the hypothesis has been made that the onset can fall in the months when children start or return to school [4,11,13]. In our NDPH patients, stressful school activity was a frequently recognized trigger event (46% of patients). Indeed, a significant increase in NDPH onset was found in the months when students started school after summer break (September) or returned to school after winter break (January) (Figure 2). 

The second most frequent trigger identified in our patients was represented by flu-like infectious episodes (22% of patients). In two previous case series [9,14], an episode of flu or a “cold” preceded NDPH onset in 30 and 42% of patients, respectively. NDPH was associated with other infective agents, such as herpes simplex virus, cytomegalovirus, salmonella, adenovirus, toxoplasma, and herpes zoster [15]. Given the variety of infectious associations, it has been suggested that the nonspecific inflammatory response initiated by the infection, rather than the infectious agent itself, may be the trigger [16]. None of these studies discuss whether, in the presence of a precipitating event, the diagnosis of NDPH can be maintained or not. It is in fact necessary to distinguish when a headache is caused by a systemic infection from when the headache is triggered by a systemic infection. According to ICHD-III (part 2, paragraph 9) when a new headache occurs for the first time in close temporal relation to an infection, it is coded as a secondary headache attributed to that infection [1].

When a pre-existing headache with the characteristics of a primary headache disorder becomes chronic or is made significantly worse (usually meaning a two-fold or greater increase in frequency and/or severity), in close temporal relation to an infection, both the initial primary headache diagnosis and a diagnosis of headache attributed to infection should be given, provided that there is good evidence that the disorder can cause headache. For headache attributed to infection, the evidence of causation should be demonstrated [1]. Headache attributed to infection is usually the consequence of active infection, resolving (acute) or not (chronic) within three months of eradication of the infection. Otherwise, NDPH triggered by an infection instead refers to a headache that persists beyond three months without evidence of another cause including an infection. We believe that the ICHD-III criteria should specify that when the headache persists for more than three months, the persistence of the infection must be demonstrated in order to differentiate a chronic infection headache from an NDPH. In our case history, only those who had a headache for more than three months with an infection resolved before were considered NDPH.

Regarding the diagnostic workout, we know that in the notes of version 3 of the ICHD, experts in particular ask to exclude other forms of headache that could mimic NDPH, for example, arteriovenous malformation, brain tumors, cerebral venous thrombosis, and chronic subdural hematoma (Table 2). 

In our cases, secondary forms of headache have been identified in 23% of patients with headache onset, suggestive for possible NDPH. All patients with secondary forms had neuroimaging or fundus oculi alterations that suggested the presence of a brain tumor, Chiari malformation type 1, or idiopathic intracranial hypertension (IIP). 

Finally, all patients with possible NDPH carried out a trial with indomethacin with a response in three patients who consequently received a diagnosis of HC. 

The clinical history of the headache, the neurological and fundus examination together with the drug history and a neuroradiological study are essential in the differential diagnosis of NDPH. Regarding the need to perform the lumbar puncture, we could raise some questions. The ICHD 2 version for the diagnosis of NDPH did not require the lumbar puncture to confirm the diagnosis of NDPH through the exclusion of alterations in the CSF pressure. The third version of ICHD includes in the notes the need to exclude in all cases of possible NDPH other headaches attributed to increased or low cerebrospinal fluid pressure. In paediatric-age patients, however, performing lumbar puncture is not always an easy procedure as it is for adults and often also requires sedation. For this reason, in our retrospective analysis, this invasive procedure was performed only in patients for whom the neurological examination or neuroimaging showed indirect signs of IIP. Patients with clinical history typical for NDPH, without papilledema and without neuroradiological alterations had not been subjected to lumbar puncture. Finally, no patients diagnosed with NDPH changed their diagnosis during follow up or developed other symptoms or signs suggestive of an alteration in CSF pressure over the time period studied. These points on the need to perform a lumbar puncture for the confirmation of NDPH should therefore be clarified.

About the outcome of patients with NDPH, we found that, in our patients, the average duration of continuous daily pain was about 8.4 ± 2 months; 43% of patients had continuous pain resolution within six months, 39% within 12 months and 18% kept suffering from continuous pain beyond 12 months. Patients who did not receive pharmacological therapy were at greater risk of having a worse prognosis—in particular, the persistence of NDPH after 12 months. Another negative prognostic indicator was the absence of a well-recognized trigger factor.

Studies in paediatric [17,18] and adult [19,20,21,22,23] NDPH patients often reached conflicting results concerning the outcome. While, according to some studies, NDPH does not interfere with the subject’s activities and its resolution is spontaneous within 24 months [22,23], others emphasize its disabling trend, drug resistance and poor prognosis [19,20,21]. 

NDPH treatments is usually a challenge. Although there are no prospective placebo-controlled trials for NDPH pharmacological treatment, the same preventive medications indicated for chronic tension-type headache or chronic migraine, including tricyclic antidepressants (amitriptyline) and antiepileptics (topiramate, valproic acid, gabapentin), are commonly used [18]. Botulinum toxin proved useful for the treatment of NDPH in adults, while no evidence is available for childhood cases [24]. Muscle relaxants, such as baclofen or tizanidine, may be helpful [20]. There are very scattered data concerning only adult populations about the use of corticosteroids [25], lidocaine [26], nimodipine [8], and ketamine [27] in NDPH. For some patients, headache escalations may respond to triptans [6]. In our series, 35% of patients did not have a response to the symptomatic drugs (NSAIDs or triptans). Regarding pharmacological treatment, around 80% of patients had a beneficial response (reduction in the frequency of attacks by at least 50%), while 20% of patients showed no improvement. The most-used drugs included flunarizine (5 mg/die) (3.7%), topiramate (1–2 mg/kg/die) (14.8%), and amitriptyline (1 mg/kg/die) (92.3%). We often chose amitriptyline for its available evidence of efficacy in chronic paediatric headaches [28]. 

Although the difficulty in finding effective therapies could lead to the development of MOH, there is no evidence of a correlation between MOH and NDPH in paediatric-age patients. Interestingly, for one of our patients, who had a concomitant history of MOH, the withdrawal of the overused drugs did not improve headache. In light of these data, and considering that even the self-limiting forms of NDPH can be disabling, we suggest trying a pharmacological approach.

### Limitations of the Study

The main limitation of this study is the retrospective nature of the data collection.

It is not possible to draw definitive conclusions on the efficacy of traditional drugs for the prophylaxis of migraine, as a comparison with a control group is not foreseen and the analysis is not longitudinal.

Not all of our patients then performed the lumbar puncture as indicated in the notes of the ICHD-III.

However, the number of patients recruited considering the paediatric age is not low, and our results offer the opportunity to extend the studies on the mechanisms of the onset of NDPH, what affects the different outcomes, and what may be the best treatment strategy.

## 5. Conclusions

NDPH represents a challenge to neuropediatricians in terms of both its diagnosis and therapy. A correct diagnosis with the exclusion of secondary causes and other chronic forms of primary headache is mandatory. Patients should be questioned about the mode of onset and the presence of trigger factors. The ICHD-III criteria should specify that when the headache persists for more than three months, the persistence of the infection must also be demonstrated in order to differentiate a chronic infection headache from an NDPH.

NDPH can last several months and be extremely disabling for the patient. Therefore, pharmaceutical treatment should be attempted, in spite of the lack of standardized guidelines for the use of drugs. In our patients, starting a treatment was associated with a more favourable outcome.

## Figures and Tables

**Figure 1 diagnostics-11-00385-f001:**
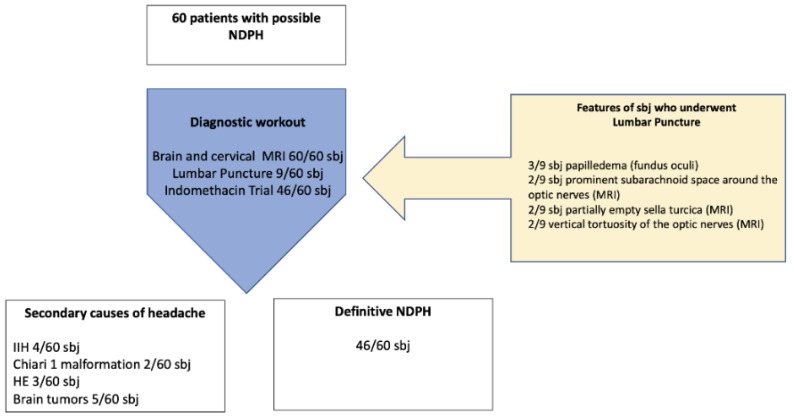
Subjects recruitment. sbj: subjects, IH: idiopathic intracranial hypertension, HC: hemicrania continua.

**Figure 2 diagnostics-11-00385-f002:**
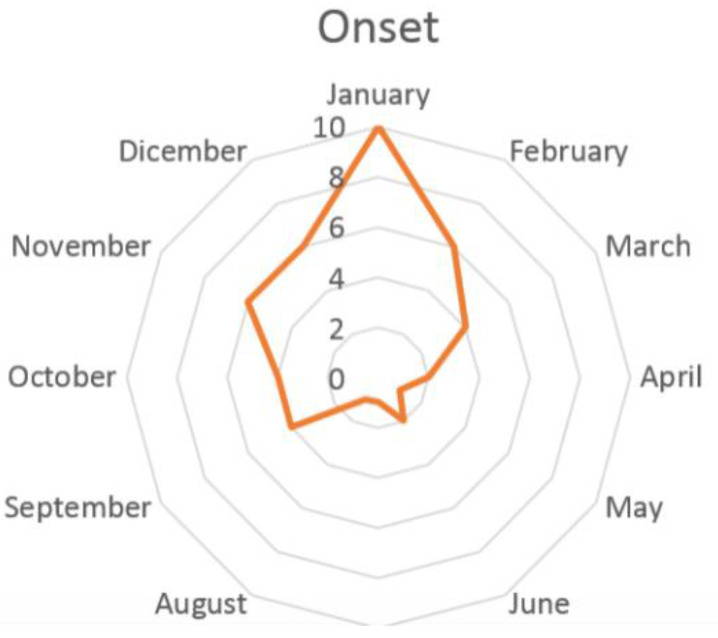
Distribution of the onset of NDPH during the year.

**Figure 3 diagnostics-11-00385-f003:**
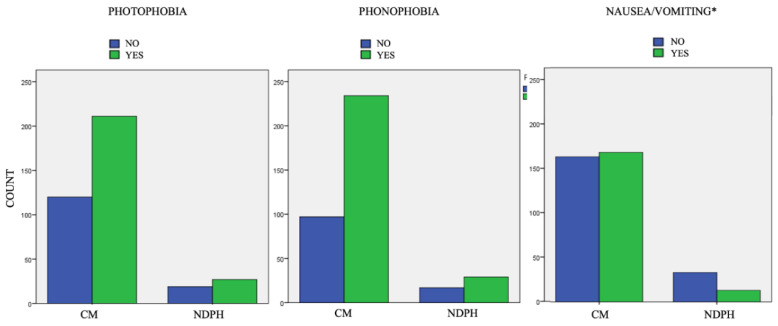
Associated symptoms in patients with new daily persistent headache (NDPH) vs. chronic migraine (CM). * *p* < 0.01.

**Table 1 diagnostics-11-00385-t001:** Location, quality, and intensity of pain in paediatric patients with NDPH.

Location of pain	Frontal	16/46 pts (35%)
	Occipital	2/46 pts (4%)
	Whole hemisphere	10/46 pts (21%)
	Monolateral	28/46 pts (61%)
	Bilateral	18/46 pts (39 %)
Intensity of pain	Mild	11/46 pts (24%)
	Moderate	23/46 pts (50%)
	Severe	12/46 pts (26%)
Pain Quality	Throbbing	19/46 pts (41%)
	Gravative	7/46 pts (15%)
	Pressing	13/46 pts (28%)
	Other quality	4/46 pts (9%)
	Not determined	3/46 pts (7%)

pts: patients.

**Table 2 diagnostics-11-00385-t002:** Differential diagnosis of NDPH.

Item	Presence of:	Consider Other Conditions:
Headache History	Previous history of primary headache Absence of trigger factor	Worsening of primary headache
	Excessive use of drugs for the attacks	Considering MOH
	Head or neck trauma	Headache secondary to trauma
	Worsening with Valsalva, or changes in posture	Altered CSF pressure
	Pain lasting 15 to 180 min	Cluster headache
	Pain lasting 2 to 30 min	Paroxysmal hemicrania
Neurological Exam	Focal sign Altered consciousness	Secondary cause of headache (i.e., vascular disorders, altered CSF pressure, neoplasia, CNS infections)
General examination	Fever	Secondary cause of headache (i.e., CSN infections)
	Prominent cranial parasympathetic autonomic features	TACs
Fundus oculi	Papilledema	Idiopathic intracranial hypertension Other secondary causes of headache.
Drug Response	Indomethacin *	Hemicrania Continua

MOH: Medication Overuse Headache; CSN: central nervous system; TACs: Trigeminal Autonomic Cephalalgias; CSF: Cerebrospinal fluid. * The indomethacin dosage necessary for successful treatment ranges from 25 to 300 mg per day, with an average of 100 mg per day. The beneficial effects appear within two days (range, one to five days). On discontinuation, headache reappears in about three days (range, 1 to 14 days).

## Data Availability

The data that support the findings of this study are available from the corresponding author, M.V., upon reasonable request.

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
