# Peer review of "Features and Management of New Daily Persistent Headache in Developmental-Age Patients"

_diagnostics, 2021, doi:10.3390/diagnostics11030385_

Round 1
Reviewer 1 Report
In mine review I wrote about the need to explain the task thatappeared at work, namely about patients whose diagnosis was modified
in relation to the one used during the visit in outpatient clinic.
As for the explanation regarding the lumbar puncture -
I would resign from the statement that its performance in
children is not easy, it is a general comment and can be applied
to any procedure that should be performed in children.
Author Response
we reported in the first paragraph of the results section that 15% of patients initially received a diagnosis other than NDPH. We have also explained the reason for this.
We changed the sentences referring to lumbar puncture in children as suggested by the reviewer.
Changes are underlined in red.

Reviewer 2 Report
I have any further comments
Author Response
Thanks.
This manuscript is a resubmission of an earlier submission. The following is a list of the peer review reports and author responses from that submission.
Round 1
Reviewer 1 Report
The paper is a retrospective one based on the patients' records from one Headache Center. According to ICHD-III criteria 46 children were diagnosed, retrospectively, to suffer from new daily persistent headaches. As the Authors state in the Material and method section, the diagnosis was re-evaluated, which means that the primary diagnosis was different than the final, made for the research. It would be interesting to find how much the primary and the final diagnosis differed from one another and what was the main field of the differences. In the same section, the reader finds information on the division of the study group into subgroups according to children's age. The description of each age group, in the context of the pain features, would be very educational especially if we consider the youngest children ( below the age of 6 years). It would be also interesting to know what was the follow-up in the presented children, how long did it last, were there any patients seen just once and diagnosed with NDPH?
In the Results section, the reader can find information about 9 out of 46 children who underwent lumbar puncture for cerebrospinal fluid pressure measures. The Authors state in Limitations of the study that the procedure of lumbar puncture was not done in each patient and, indeed, it is in contradiction with Notes( see ICHD-III, point 4.10 New daily persistent headache, Notes: In all cases other secondary headaches...should be ruled out by appropriate investigations). The main concern, from my point of view, is to prove that the diagnosis of NDPH was properly done without cerebrospinal fluid pressure measure.
In the section Treatment, one can read about prophylaxis with amitriptyline, flunarizine, or topiramate. However- NDPH criteria state that the pain must be persistent and daily from its onset ( previously- chronic headache with acute onset) so my linguistic and logical doubt is if, then, one can call the treatment" prophylaxis" in this case.
Reviewer 2 Report
The authors included 380 pediatric patients with chronic headache. The prevalence of NDPH was 15%. Almost 58% of NDPH patients received a prophylactic therapy and the most used drug was amitriptyline. A reduction of attacks by at least 50% in a month was detected in 30.6% of patients.
The authors concluded that although prophylaxis is recommended, considering the long duration that this headache can have, there are not data supporting the treatment choice.
Comments:
- More precisely, an indotest should be performed to rule out not only Hemicrania continua but also paroxysmal hemicrania
- Statistical analysis should be revised by an expert. The authors performed correlations that not could be defined as correlations (e.g. NDPH and sex???) and many analysis with a significant increase of type I error. Anyway, this risk should be reported in the limits section.
- Authors say that patients should be questioned about the mode of onset and the presence of trigger factors. The authors should stress that there is a relevant diferrence between referred triggers and triggers reported after a survey. This is true in adults and maybe it is more important in pediatrics. Please stress this point in the discussion/conclusions. See and cite other relevant papers about trigger management and variation in migraine/headache : Baldacci F, et al. How aware are migraineurs of their triggers? Headache. 2013 May;53(5):834-7; Yamanaka G, et al. A Review on the Triggers of Pediatric Migraine with the Aim of Improving Headache Education. J Clin Med. 2020 Nov 19;9(11):3717; Baldacci F, et al. Triggers in allodynic and non-allodynic migraineurs. A clinic setting study. Headache. 2013 Jan;53(1):152-160. Tai MS, et al. Geographical Differences in Trigger Factors of Tension-Type Headaches and Migraines. Curr Pain Headache Rep. 2019 Feb 21;23(2):12.)
- The authors reported in the text that: “All 60 patients with possible NDPH underwent brain computed tomography (CT) or magnetic 105 resonance imaging (MRI) to exclude secondary cerebral causes.” This is not fit with figure 1 and material and methods as well: “Secondary headaches were appropriately excluded in all patients based on…contrast-enhanced magnetic resonance imaging”. Anyway, they should report in the limits section that realistically only a part of these patiens performed brain MRI and MRA with gadolinium. They also should add that psychiatric scales were not systematically used in the assessment
- In the conclusion i would underline the reprisal of the discussion “We believe that the ICHD-III criteria should specify that when the headache persists for more than 3 months, the persistence of the infection must be demonstrated in order to differentiate a chronic infection headache from an NDPH”
Reviewer 3 Report
This is an interesting study because of the novelty and the strange of the treated disorder. However, in my opinion, the present study does not meet the necessary requirements to be published.
- The title does not reflect the reality of the study. Authors have included not only patient with New Daily Persistent Headache (NDPH), but also with chronic migraine (CM), chronic tension-type headache (CTTH) or undefined headache, being the latter more than the 85% of the total sample. In addition, authors say that the study is performed in a pediatric sample, but authors also expose, in line 56, that the data “was searched for all children and adolescents up to 18 years of age”.
- Authors inform in abstract section that the total sample of the study is 380 subjects, but being consistent with the study aim, exposed in lines 13-14 and 51-52, the total sample with NDPH is formed by 46 subjects.
- The introduction section does not put the reader into context correctly. The text is almost copy paste from “Headache Classification Committee of the International Headache Society (IHS) The International Classification of Headache Disorders, 3rd edition. 2018 Jan;38(1):1-211. doi: 10.1177/0333102417738202”.
- Authors are not clear describing the characteristic of their sample, generating a lot of doubts to understand data such as the total sample and its characteristics Author describe between lines 64-65 “primary headaches, including chronic migraine, chronic tension-type headache, and hemicrania continua other than NDPH, were ruled out”. In spite of the above, between lines 99-100, authors also describe that “other chronic primary headaches in our sample were chronic migraine (CM, 65%), chronic tension 100 type headache (CTTH, 12%)”.
- I have an important doubt in line to the diagnosis, the first rule to stablish a headache diagnosis is that the diagnosis should not bet better accounted for by another ICHD-3 diagnosis. Initially, author expose: “We retrospectively reviewed the charts of patients attending to the tertiary, pediatric medical Headache Centers of Bambino Gesù Children Hospital”. In this point, I supouse that patient have a previous headache diagnosis. Then, authors expose: “The diagnosis was reevaluated in all cases by using the ICHD-III criteria”. My doubt appears in this point, how new NDPH are identified from a previous diagnosis? Are previous diagnoses not valid? Why are patients, at the begging, diagnosed of another type of headache and now it is believed that it could be NDPH, if data are collected from a data base (line 55-57)?
- Maybe as cause of the ambiguous description of the sample and its characteristics, I have important doubts about the NDPH diagnosis of some subjects of the sample. Attending to the qualitative features and the associated symptoms of the NDPH sample exposed in results, I think that patient are better accounted in a CM or CTTH diagnosis
- In results section, the information contained between lines 96 and 119 must be included in methods section. Between these lines, authors make an attempt to explain the selection process of the sample.
- Authors, in lines 96-97 have informed of the mean age of the possible eligible sample (390 subjects), but the interesting data is mean age of the 46 NDPH patients, and its not appear in the manuscript.
- In lines 86-87 authors write: “Statistical analysis was conducted by SPPS version 22.0 and Ó˝2 test was used to verify possible correlations between: (a) NDPH and population features (age and sex)”. Age is a continuous variable and Ó˝2 test is recommended to verify correlation between two dichotomic variables. This analysis is not well performed.
- I do not reach to understand the sense to perform an univariate logistic regression, what is the purpose of this analysis? Which variables are involved in this analysis? Where is possible to observe the results of this analysis?
- I miss a table 1 where sample characteristics be well explained.
- Discussion section seems more an extensive introduction section than a discussion section.